# Cross-sectional study to describe allergic rhinitis flare-ups and associated airways phenotype in house dust mite sensitization

**Ludovic de Gabory[1], Sabine Amet[2]\*, Annelore Le Maux[2], Jean-Pierre Meunier[3], Antoine Chartier[2], Cécile Chenivesse[4]**

**1** Otorhinolaryngology Department, Hôpital Pellegrin, University Hospital of Bordeaux, Bordeaux, France, **2** Direction Médicale, Allergologisk Laboratorium København Société par Actions Simplifiées (ALK SAS), La Défense, France, **3** Direction Médicale, Axonal, Nanterre, France, **4** Service de Pneumologie et Immuno-Allergologie, CHU de Lille, Univ. Lille, CNRS, Inserm, Institut Pasteur de Lille U1019 - UMR 9017 - CIIL - Center for Infection and Immunity of Lille, CRISALIS / F-CRIN INSERM Network, Lille, France

\* sabine.amet@alk.net

**Data Availability Statement:** All relevant data are within the manuscript and its Supporting Information files.

## Abstract

### Objectives

To quantify and describe flare-ups of house dust mite allergic rhinitis (HDM-AR) which had occurred during the last 12 months in a population of adults and children candidate for Allergen ImmunoTherapy (AIT). Next, to identify associated clinical features.

### Materials and methods

This was an observational, multicenter, cross-sectional study that included patients aged ≥ 5 years with HDM-AR eligible for AIT and without prior AIT for at least 12 months. Flare-ups were all period with impairment of quality of life (QoL) and requiring a change in their usual treatment. Data were collected using medical records and patient questionnaires. Variables associated with the occurrence of ≥ 2 AR flare-ups were identified.

### Results

1,701 patients were included (average age: 23 years, 51.5% males, 30.4% children, 17.7% adolescents and 51.9% adults). Severe and persistent AR affected 70.9% of them and 53.7% showed polysensitization. Asthma was associated with AR in 34.4% and was well-controlled in 58.5%. The occurrence of at least one AR flare-up in the year was reported by 77.7%, with an annual rate in the whole population of 2.6 ± 3.9 and a duration of 14.1 ± 17.1 days. Deeply or moderately AR-related degraded QoL was experienced by 39.5% and 64.6%, respectively. The occurrence of ≥ 2 AR flare-ups was reported by 54.5% and was associated with polysensitization, AR intermittence and severity.

### Conclusion

AR flare-ups are frequent and impair QoL in HDM-allergic patients, suggesting that it could be considered as therapeutic targets.

**Funding:** This study is funded by Allergologisk Laboratorium København Société par Actions Simplifiées (ALK SAS). The study design, data collection and analysis, and the preparation of the manuscript were provided by Axonal-biostatem, Nanterre, France.

**Competing interests:** Ludovic de Gabory: Honoraria/Consulting: ALK-SAS, AstraZeneca, Chiesi, GlaxoSmithKline, Integra Life Science, Laboratoire de la Mer, Laboratoire Chemineau, Medtronic, Sanofi Genzyme, Zambon. Sabine Amet: ALK SAS corporate employee Annelore Le Maux: ALK SAS corporate employee Jean-Pierre Meunier: Manager of the CRO Axonal, which was designated by ALK to conduct the trial Antoine Chartier: ALK SAS corporate employee Cécile Chenivesse: Grants from AstraZeneca, Santelys Personal fees from ALK SAS, AstraZeneca, Boehringer Ingelheim, Chiesi, GlaxoSmithKline, Novartis, Sanofi-Regeneron, TEVA Congress support from ALK SAS, AstraZeneca, Boehringer Ingelheim, GlaxoSmithKlein, Novartis, Pierre Fabre, Pfizer, Roche, TEVA

## Introduction

The worldwide prevalence of allergic rhinitis (AR) is around 10 to 20% and mostly caused by house dust mites (HDM) [1–3]. It is often associated with asthma [4, 5] and both allergic reactions have their own medical history and interactions [6, 7].

Although it is well known that asthma can be uncontrolled, leading to emergency visits, hospitalizations, intensive care and unfortunately sometimes death, AR is perceived as a mild benign disease without hospitalization and death. However, AR patients present impaired quality of life (QoL) to the same extent as asthma patients [8–10]. While asthma therapeutic strategy is primarily aimed at achieving control of symptoms and reducing number of exacerbations, the AR strategy is to control symptoms and improve QoL [11].

Until now, it is difficult to know whether in the history of AR, flare-ups are observable in the same way as for asthma notably when the former is persistent and severe. Moreover, potential AR flare-ups are difficult to quantify because they do not lead to urgent medical action, an indicator of care. However, these events would be considered to be part of AR management. Recently, Demoly *et al* addressed the notion of "disease control" in AR combining the measurements of the severity and/or frequency of diurnal or nocturnal symptoms, the impairments in social, physical, professional and educational activities, respiratory function monitoring and exacerbations [12]. As in the field of asthma, we hypothesized that AR flare-ups occurred in perennial HDM allergy and could be related to the importance of exposure and associated factors.

The main purpose of this study was to quantify and clinically describe AR flare-ups which had occurred during the last 12 months in a population of adults and children's candidates for an HDM-Allergen ImmunoTherapy (AIT). Our first hypothesis is that there are flare-ups in AR as well as in asthma and that flare-ups might follow the seasonality and the epidemiology of the mites [13].

## Materials and methods

We conducted a multicenter and cross-sectional study which was proposed to a representative sample of 200 allergists in France between September 2017 and May 2018, under real-life medical conditions. Patients attended a single visit at inclusion, during which main characteristics and clinical symptoms of AR and asthma were reviewed by the physician. Patients (or the parents for minor children) completed a self-questionnaire before an initial dose of AIT.

This study was carried out in accordance with the ethical principles of the declaration of Helsinki, Good Epidemiological Practice, and the reference methodology MR 003 granted by the French Data Protection Agency (CNIL). This study was registered with the French authorities under 2017-A01903-50 identification number on June 2017 and was approved by the Ethics Committee on September 2017. The physician had to obtain written consent from the patient before the inclusion in the study. For minor patients, a parent or guardian was informed and had to give written consent for the patient to participate.

Each physician included the first 12 patients with HDM-AR (clinical perennial relevant symptoms plus concordant HDM positive skin prick test and/or specific serum immunoglobulin E > 0.70 U), who were proposed initiation of AIT. Patients aged 5 and older, without other rhinitis nor chronic rhinosinusitis, with or without asthma, able to fill out a questionnaire (or with able parents for minor children), and having signed an informed consent form, were included. Exclusion criteria were patients who had received AIT for HDM during the past 12 months, those participating in an interventional study, or who refused to participate. The population included were divided into three groups based on age: children (aged 5–11 years), adolescents (12-17 years) and adults (aged 18 years and older).

The primary outcome was the description of AR flare-ups in the past 12 months. Patients were asked to remind if they had had any symptoms that were so severe that they required a change in their usual treatment and had an impact on their quality of life. AR flare-up was documented by recording their frequency, the season in which they occurred, the type and duration of AR symptoms, their influence on QoL and their worsened factors. In younger age patients, AR flare-up was defined in the same way but in this case, the count and description were reported by the parents.

Additionally, the patient profiles were described, and variables associated with frequent (≥2) AR flare-ups were explored.

The secondary outcomes were the description of AR and asthma characteristics by age, asthma exacerbation, QoL of respiratory allergy patients and patient profiles according to asthma control status.

After inclusion, the physician collected the patient's socio-demographic data, concomitant allergic diseases, comorbidities, clinical history of the rhinitis (date of onset/diagnosis, frequency and severity of symptoms according to the ARIA guidelines (2)), clinical history of the asthma and risk factors for exacerbations according to the Global Initiative for Asthma [GINA] classification [14], number of exacerbations within the last 12 months based on the international definition of asthma exacerbations [15, 16], and concomitant treatments for AR and/or asthma. Next, patients were asked to recall whether any AR flare-ups had occurred within the last 12 months, their worsened factors and their impact on QoL (very degraded, moderately degraded, unchanged, improved). To quantify symptoms and QoL impairment several validated self-questionnaire have been used: the Nasal Obstruction Symptom Evaluation score (NOSE), the first 13 questions of the DyNaChron questionnaire (loss of smell) and the RhinoQOL questionnaires (sleep, social and physical capacities, mood disorders consequences) were synergistically combined and were completed by the patient [17, 18]. Next, the adult version of the asthma control questionnaire (ACQ-6) and the interviewer-administered version of ACQ for children 6-10 years were used to measure asthma control: 6-items scored from 0 = totally controlled to 6 = severely uncontrolled; asthma patients with a score below 1.0 were considered to have adequately controlled asthma and those having a score above 1.5 were considered to be uncontrolled [19].

Given an expected frequency of AR flare-ups of around 10 to 20%, it was deemed necessary to include 1,768 to 2,079 patients in the overall population [20]. Based on an adult/adolescent/child distribution estimated from the Antares study at 60%, 20% and 20%, respectively, enrollment had to be around 1,200 adults, 400 adolescents and 400 children.[4] With these numbers, the accuracy rate for the measurement of the rate of flare-ups was 1.8 to 2.2% for the adult population, and 3.2% to 3.8% for the adolescent and child populations.

Taking into account a rate of missing or unanalyzable data of 15%, a total population of about 2,000 patients was considered sufficient to provide enough detail to describe the flare-up rate and its 95% confidence interval (CI).

The statistical analysis was performed using SAS® software (SAS Institute, NC, Cary, USA, version 9.4). Analysis was performed on the whole population. Missing data were not replaced. AR flare-ups and asthma exacerbations, as well as patient characteristics, were described by age group, and the level of control of asthma. For comparisons, the Kruskal-Wallis test was used. Logistic regression analyses were carried out to identify variables associated with ≥ 2 AR flare-ups and ≥ 2 asthma exacerbations (the threshold of 2 exacerbations was chosen in order to strengthen the diagnosis of exacerbations). The factors analyzed were age, gender, smoking habit, cannabis consumption, conjunctivitis, atopic eczema, food allergy, allergic urticaria, time from diagnosis, polysensitization, drug allergy, hymenoptera venom allergy, occupational allergy, rhinitis severity according to the ARIA guidelines [1], sneezing, rhinorrhea, nasal

obstruction (NO), nasal/ocular pruritus, loss of smell, redness and tearing of eyes, asthma, NOSE, RhinoQOL and DyNaChron scores. Finally, a multivariate analysis was built using variables that reached a p-value ≤ 0.20 with less than 20% missing data in the univariate analyses. The final model was derived from the selection of variables by a stepwise regression procedure with a threshold set at 20% for including and 5% for excluding the independent variables. From the final model, odds ratios (ORs) were presented with the 95% CI.

## Results

A total of 1701 patients were analyzed (Fig 1) and patient characteristics are presented in Table 1. The main result is that 70.9% of the whole population suffered from persistent severe AR. Moreover, HDM-AR diagnosis had been established for 3.2 ± 5.9 years. The period increased from children to adolescents and to adults (1.3±1.7, 2.6±3.3 and 4.5±7.6). The most frequent AR symptoms were rhinorrhea (96.8%), sneezing (96.4%) and NO (93.1%). (Table 1).

The total scores of self-questionnaires are showed in Table 2. The mean NOSE score was 55.4±26.7, including 804 patients (54.5%) with severe NO (score>50) (Table 2). The NOSE score was 29.4±23.6 for mild intermittent, 49.8±25.0 for severe intermittent, 41.1 ± 21.9 for mild persistent and 62.9±24.1 for severe persistent AR. Severe NO was more frequent in patients with severe, intermittent (55.2% of patients) or persistent (65.8%) AR than in patients with mild intermittent (16.9%) or persistent (29.8%) AR. The DyNaChron score was 21.9±23.6 for mild intermittent, 49.0±35.0 for severe intermittent, 31.6±25.7 for mild persistent and 57.8 ±29.3 for severe persistent AR. The NOSE, Impact score of RhinoQoL questionnaire and DyNaChron scores tended to increase with age.

Patients reported an average of 2.6±3.9 AR flare-ups in previous year (Table 3): 22.3% of patients without any AR-flare-up, 23.2% with 1, 18.2% with 2, 15.8% with 3, 5.7% with 4, 7.6% with 5 and 7.2% with more than 5. Among the 54.5% of patients who reported at least 2 AR flare-ups, the mean duration of AR flare-ups was 14.5±16.7 days (median = 10 days) and the main worsened factors were infections (37.9%) and pollution (27.7%). They occurred more frequently during the last 4 months of the year (Fig 2). AR flare-ups mostly impacted QoL with 39.5% and 64.6% of patients having experienced at least one flare-up associated with a very and moderately degraded QoL, respectively. Polysensitization, the severity and intensity of AR and NOSE score>50 were associated with ≥2 AR flare-ups (Table 4).

Asthma was reported in 34.4% of patients and had been diagnosed for 5.2±8.1 years (2.6 ±2.5 in children, 3.4±4.0 in adolescents and 8.4±11.1 in adults). There were 41% of asthmatic patients in GINA step 1, 23.6% in step 2, 27.4% in step 3, 7.8% in step 4 and 0.2% in step 5. Asthma was well controlled in 58.5% of cases, and the proportion of patients with ACQ-6 score>1.5 was 39.6% (Table 2).

The frequency of asthma exacerbations was 2.4±4.8 events a year. 40.9% of asthmatic patients had no asthma exacerbation during the previous year, and 59.1 and 43.8% had at least 1 or 2 asthma exacerbations, respectively. The average number of severe exacerbations was 0.5±1.3 (median = 0) with 26.1% of asthmatic patients requiring at least one burst of oral corticosteroids and 3.7%, hospitalization. Variables associated with the occurrence of ≥2 exacerbations of asthma were control level, atopic eczema, loss of smell and stage 2 to 4-5 of the GINA classification (Table 4).

The HDM-AR characteristics according to control asthma are described in Table 5. Among well controlled asthma there was 64.7% of severe AR compared to 72.5% and 74.5% in the partly and poorly controlled subgroups. Compared to patients with well controlled asthma, patients with partly or poorly controlled asthma had higher NOSE score and seemed to have a higher RhinoQOL impact score. The mean number of AR flare-ups was 2.3±2.3 for well controlled patients, 2.4±2.4 for those partly controlled and 5.2±9.8 for those poorly controlled.

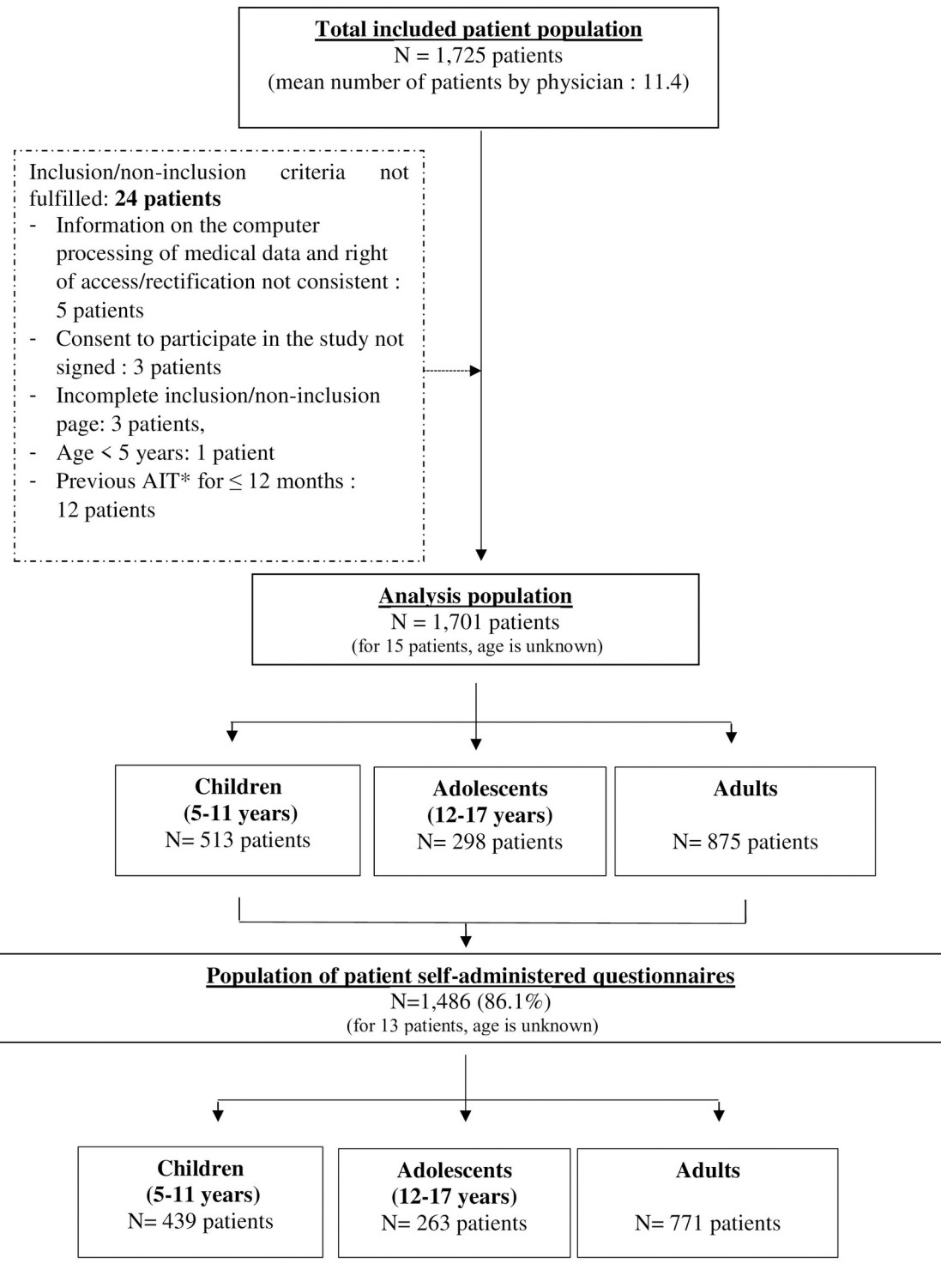

Flow chart of study population
*Allergen Immuno Therapy

**Fig 1. Flow chart of study population.**

**Table 1. Characteristics of study population.**

| | | | 5-11 years | 12-17 years | Adults | Total* |
|---|---|---|---|---|---|---|
| | | | **N = 513** | **N = 298** | **N = 875** | **N = 1,701** |
| **Gender** | | N[1] (missing) | 513 (0) | 297 (1) | 875 (0) | 1,694 (7) |
| | Male | N[2] (%) | 330 (64.3%) | 168 (56.6%) | 370 (42.3%) | 873 (51.5%) |
| **Age (years)** | | N (missing) | 513 (0) | 298 (0) | 875 (0) | 1,686 (15) |
| | | Mean ± SD | 8.4 ± 1.9 | 14.2 ± 1.7 | 34.6 ± 12.4 | 23.0 ± 15.1 |
| **Smokers** | | N (missing) | 512 (1) | 296 (2) | 872 (3) | 1,690 (11) |
| | | n (%) | 8 (1.6%) | 10 (3.4%) | 100 (11.5%) | 119 (7.0%) |
| **Polysensitization** | | N (missing) | 511 (2) | 294 (4) | 863 (12) | 1,679 (22) |
| | | n (%) | 241 (47.2%) | 156 (53.1%) | 498 (57.7%) | 902 (53.7%) |
| **Allergic comorbidities** | | N | 513 | 298 | 875 | 1,701 |
| | Conjunctivitis | n (%) | 249 (48.5%) | 163 (54.7%) | 529 (60.5%) | 954 (56.1%) |
| | Atopic eczema | n (%) | 132 (25.7%) | 62 (20.8%) | 155 (17.7%) | 354 (20.8%) |
| | Food allergy | n (%) | 40 (7.8%) | 28 (9.4%) | 72 (8.2%) | 144 (8.5%) |
| **Severity of AR** | | N (missing) | 484 (29) | 277 (21) | 823 (52) | 1,599 (102) |
| | Mild intermittent | n (%) | 56 (11.6%) | 25 (9.0%) | 57 (6.9%) | 139 (8.7%) |
| | Severe intermittent | n (%) | 8 (1.7%) | 2 (0.7%) | 23 (2.8%) | 33 (2.1%) |
| | Mild persistent | n (%) | 114 (23.6%) | 46 (16.6%) | 130 (15.8%) | 293 (18.3%) |
| | Severe persistent | n (%) | 306 (63.2%) | 204 (73.6%) | 613 (74.5%) | 1,134 (70.9%) |
| **Asthma** | | N (missing) | 505 (8) | 288 (10) | 854 (21) | 1,661 (40) |
| | | n (%) | 215 (42.6%) | 106 (36.8%) | 248 (29.0%) | 572 (34.4%) |

* The total is different from the sum of the three categories because age was unknown for 15 patients

1 Number of observations filled in

2 Number of patients

**Table 2. Results from self-questionnaires.**

| *All patients* | | | 5-11 years | 12-17 years | Adults | Total* |
|---|---|---|---|---|---|---|
| | | | **N = 439** | **N = 263** | **N = 771** | **N = 1,486** |
| **NOSE (0-100)** | | N (missing) | 436 (3) | 260 (3) | 767 (4) | 1,476 (10) |
| | | Mean ± SD | 50.9 ± 25.8 | 56.4 ± 26.0 | 57.7 ± 27.1 | 55.4 ± 26.7 |
| **RHINOQOL (0-100)** | Frequency | N (missing) | 420 (19) | 248 (15) | 735 (36) | 1,415 (71) |
| | | Mean ± SD | 65.8 ± 20.6 | 60.7 ± 19.8 | 59.6 ± 20.9 | 61.6 ± 20.8 |
| | Bothersomeness | N (missing) | 360 (79) | 229 (34) | 684 (87) | 1,284 (202) |
| | | Mean ± SD | 69.0 ± 20.8 | 61.3 ± 20.5 | 58.8 ± 22.9 | 62.1 ± 22.4 |
| | Impact | N (missing) | 425 (14) | 256 (7) | 755 (16) | 1,447 (39) |
| | | Mean ± SD | 21.6 ± 19.0 | 26.8 ± 20.4 | 32.1 ± 21.7 | 28.1 ± 21.2 |
| **DYNACHRON (0-130)** | | N (missing) | 424 (15) | 251 (12) | 745 (26) | 1,432 (54) |
| | | Mean ± SD | 41.9 ± 28.3 | 51.9 ± 30.8 | 52.9 ± 32.4 | 49.4 ± 31.3 |
| *Asthma patients* | | | | **N = 91** | **N = 212** | **N = 303** |
| **ACQ-6 (0-6)** | | N (missing) | | 84 (7) | 201 (11) | 285 (18) |
| | | Mean ± SD | | 1.2 ± 1.0 | 1.5 ± 1.2 | 1.4 ± 1.1 |

* The total is different from the sum of the three categories because age was unknown for 13 patients

**Table 3. Description of AR flare-ups by age group in the last 12 months according to the patient's opinion.**

| | | | 5-11 years | 12-17 years | Adults | Total* |
|---|---|---|---|---|---|---|
| | | | N = 439 | N = 263 | N = 771 | N = 1,486 |
| **AR flare-up** | | N (missing) | 337 (102) | 196 (67) | 623 (148) | 1,166 (320) |
| | | Mean ± SD | 2.4 ± 3.5 | 2.3 ± 3.1 | 2.8 ± 4.2 | 2.6 ± 3.9 |
| | | < 2 | 146 (43.3%) | 96 (49.0%) | 285 (45.8%) | 530 (45.5%) |
| | | ≥ 2 | 191 (56.7%) | 100 (51.0%) | 338 (54.2%) | 636 (54.5%) |
| **Additional treatments** | | N (missing) | 364 (75) | 206 (57) | 645 (126) | 1228 (258) |
| | | % | 82.9 | 78.3 | 83.7 | 82.6 |
| *In patients with at least 2 AR flare-ups* | | | | | | |
| **Duration** | | N (missing) | 180 (11) | 87 (13) | 319 (38) | 592 (44) |
| | | Mean ± SD | 12.7 ± 12.3 | 13.6 ± 15.2 | 15.6 ± 19.1 | 14.5 ± 16.7 |
| **Symptoms** | | N | 191 | 100 | 338 | 636 |
| | Sneezing | n** (%) | 161 (84.3%) | 91 (91.0%) | 287 (84.9%) | 545 (85.7%) |
| | Rhinorrhea | n (%) | 156 (81.7%) | 90 (90.0%) | 288 (85.2%) | 541 (85.1%) |
| | Nasal obstruction | n (%) | 162 (84.8%) | 90 (90.0%) | 281 (83.1%) | 540 (84.9%) |
| | Conjunctivitis | n (%) | 75 (39.3%) | 46 (46.0%) | 145 (42.9%) | 267 (42.0%) |
| | Loss of smell | n (%) | 32 (16.8%) | 30 (30.0%) | 131 (38.8%) | 195 (30.7%) |
| **Aggravating factors** | | N | 191 | 100 | 338 | 636 |
| | Infection | n (%) | 79 (41.4%) | 33 (33.0%) | 127 (37.6%) | 241 (37.9%) |
| | Pollution | n (%) | 39 (20.4%) | 28 (28.0%) | 106 (31.4%) | 176 (27.7%) |
| | Stress | n (%) | 14 (7.3%) | 15 (15.0%) | 96 (28.4%) | 126 (19.8%) |
| | Tobacco | n (%) | 4 (2.1%) | 8 (8.0%) | 34 (10.1%) | 47 (7.4%) |
| | Professional exposure | n (%) | 1 (0.5%) | 3 (3.0%) | 30 (8.9%) | 34 (5.3%) |
| **Impact on QoL** | | N | 191 | 100 | 338 | 636 |
| | Very degraded | n (%) | 65 (34.0%) | 32 (32.0%) | 152 (45.0%) | 251 (39.5%) |
| | Moderately degraded | n (%) | 114 (59.7%) | 65 (65.0%) | 225 (66.6%) | 411 (64.6%) |
| | Unchanged | n (%) | 42 (22.0%) | 19 (19.0%) | 37 (11.0%) | 98 (15.4%) |
| | Improved | n (%) | 10 (5.2%) | 3 (3.0%) | 11 (3.3%) | 25 (3.9%) |

* The total is different from the sum of the three categories because age was unknown for 13 patients

** Number of patients with at least one impact

## Discussion

The results of this study support the existence of AR flare-ups in patients with HDM-AR. The phenotype "frequent exacerbator" defined as having 2 or more AR annual flare-ups was associated with polysensitization, AR severity and intermittence and severe NO.

Our population reflected the general population suffering from HDM-AR and eligible to AIT: 100% perennial, 70.9% of persistent severe AR, 50.7% polysensitized patients, 34.4% associated asthma [4, 21]. In our study NO due to AR was able to reach the level of symptom comparable to structural cause due to nasal septal deviation [22, 23]. It is well known today that NO, whatever the cause, is an etiology of sleep disorders, contributing to a large part of the AR burden [8, 10]. These results highlight the level of QoL impairment due to AR. Moreover, the scores of the DyNaChron questionnaire [18], showed that the sense of smell was altered and participated in QoL impairment with varying feelings depending on the age group (slightly greater in adolescents and adults). However, this score was less altered in our population than in other populations of patients suffering from chronic rhinosinusitis with or without polyps [24].

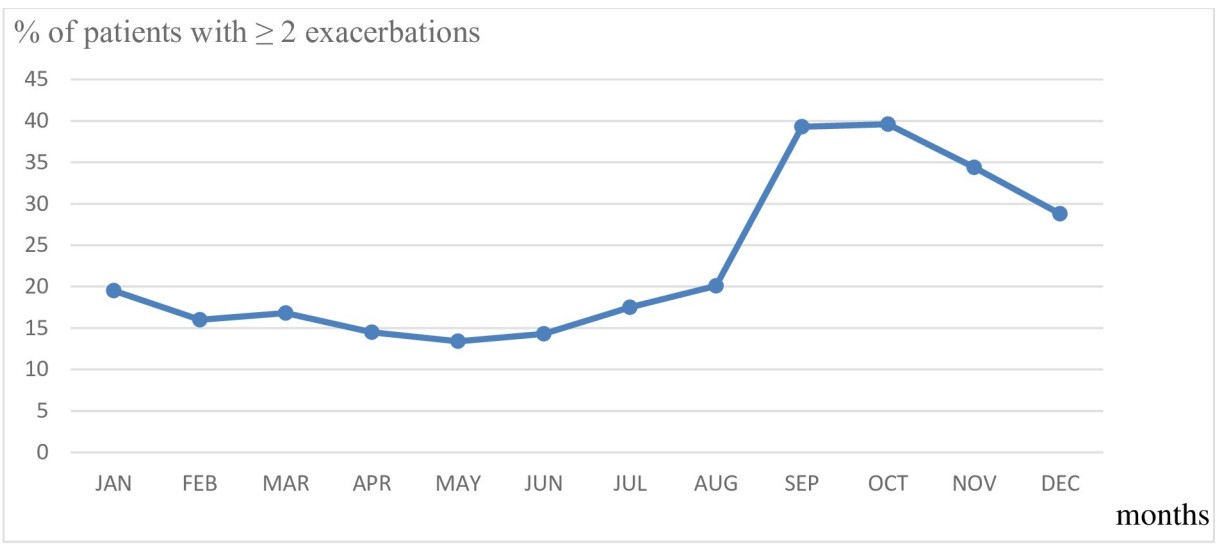

% of patients with ≥ 2 exacerbations

*This curve describes the percentage of patients with at least 2 exacerbations over the past 12-months.*

**Fig 2. Frequency of AF flare-ups by month.**

Flare-ups during AR are poorly described in the literature. Nowadays, they have no consensus definition [25] and symptoms are not objectively measurable with reliable instrumentation except with self-questionnaires [26, 27] Randomized clinical trials of 12 SQ-HDM AIT in AR used to define a rhinitis flare-up as a day when the subject returned to the high level of symptoms required for trial inclusion. Data collected at randomization were used to characterize rhinitis flare-ups [20]. As we could not use this definition in a real-life survey, we chose to define AR flare-ups in agreement with a recent systematic review reporting that the common point of all definitions used for AR flare-ups was the worsening of chronic rhinosinusitis symptoms with a return to baseline symptoms following treatment [25]. In order to facilitate recall and overcome the daily variation of symptoms, we added the notion of impaired quality of life and the need to modify usual treatment. We found a prevalence around 2.6 ± 3.9 in the previous 12 months, especially in adults and adolescents and lasting 14.1 ± 17.1 days, apparently longer in adults and adolescents. Our results are consistent with a recent publication by *Jason H Kwah and al* that found 19.3% of patients with chronic rhinosinusitis experiencing more than 4 flare-ups requiring antibiotic per year based on a data collected from an electronical medical record (20.5% in our study) [28]. We showed also that AR flare-ups occurred mostly in autumn, regardless of age. This could be explained by HDM sensitization. Allergy to HDM is perennial with 2 seasonal increases, one during fall and one during spring [13].

Polysensitization and NOSE >50 were strong statistical indicators of flare-ups for AR and could be an early clinical indicator to start AIT. Controlling AR flare-ups seems all the more important since it is usually considered that when the nose is controlled the bronchi are often easier to control [7]. Moreover, uncontrolled allergic rhinitis has been shown to impair the results of nasal septum surgery when two obstructive factors coexist in the nasal cavity before treatment [23]. AR flare ups should prioritize patients requiring increased treatment, more therapeutic education, more frequent visits and earlier desensitization. Finally, international [1] and French guidelines [29] recommend initiation of AIT when symptomatic treatments are insufficient to control symptoms and when patient QoL is very degraded. Our study

**Table 4. Variables associated with the occurrence of 2 exacerbations of AR and asthma.**

| AR | | Total | < 2 exacerbations | ≥ 2 exacerbations | Univariate analysis | | Multivariate analysis | |
|---|---|---|---|---|---|---|---|---|
| | | N = 1,166 | N = 530 | N = 636 | p-value* | OR | p-value** | OR |
| **Polysensitization** | Missing | 14 | 6 | 8 | | | | |
| | No | 533 | 274 (51.4%) | 259 (48.6%) | <0.001 | Ref | 0.001 | Ref |
| | Yes | 619 | 250 (40.4%) | 369 (59.6%) | | 1.56 [1.24 ; 1.97] | | 1.57 [1.19 ; 2.07] |
| **ARIA Classification** | Missing | 65 | 25 | 40 | | | | |
| | Mild persistent | 200 | 125 (62.5%) | 75 (37.5%) | <0.001 | Ref | 0.002 | Ref |
| | Mild intermittent | 100 | 53 (53.0%) | 47 (47.0%) | | 1.48 [0.91 ; 2.40] | | 1.77 [1.02 ; 3.10] |
| | Severe Persistent | 777 | 320 (41.1%) | 457 (58.2%) | | 2.38 [1.73 ; 3.28] | | 1.91 [1.30 ; 2.80] |
| | Severe Intermittent | 24 | 7 (29.2%) | 17 (70.8%) | | 4.05 [1.60 ; 10.21] | | 5.34 [1.66 ; 17.11] |
| **NOSE score** | Missing | 7 | 2 | 5 | | | | |
| | ≤ 50 | 523 | 295 (56.4%) | 228 (43.6%) | <0.001 | Ref | <0.001 | Ref |
| | > 50 | 636 | 233 (36.6%) | 403 (63.4%) | | 2.24 [1.77 ; 2.84] | | 1.92 [1.43 ; 2.57] |

| ASTHMA | | Total | < 2 exacerbations | ≥ 2 exacerbations | Univariate analysis | | Multivariate analysis | |
|---|---|---|---|---|---|---|---|---|
| | | N = 443 | N = 249 | N = 194 | p-value* | OR | p-value** | OR |
| **Control** | Missing | 80 | 51 | 29 | | | | |
| | Well controlled | 214 | 145 (67.8%) | 69 (32.2%) | <0.001 * | Ref | 0.002 | Ref |
| | Partly controlled | 94 | 38 (40.4%) | 56 (59.6%) | | 3.10 [1.87 ; 5.12] | | 2.36 [1.29 ; 4.28] |
| | Poorly controlled | 55 | 15 (27.3%) | 40 (72.7%) | | 5.60 [2.90 ; 10.83] | | 3.22 [1.39 ; 7.05] |
| **Atopic eczema** | No | 333 | 203 (61.0%) | 130 (39.0%) | <0.001 * | Ref | 0.001 | Ref |
| | Yes | 110 | 46 (41.8%) | 64 (58.2%) | | 2.17 [1.40 ; 3.37] | | 2.63 [1.39 ; 7.05] |
| **Loss of smell** | No | 239 | 152 (63.6%) | 87 (36.4%) | <0.001 * | Ref | 0.008 | Ref |
| | Yes | 204 | 97 (47.5%) | 107 (52.5%) | | 1.93 [1.32 ; 2.82] | | 2.03 [1.20 ; 3.43] |
| **GINA classification** | Missing | 6 | 6 | 0 | | | | |
| | Level 1 | 167 | 118 (70.7%) | 49 (29.3%) | <0.001 * | Ref | 0.012 | Ref |
| | Level 2 | 102 | 54 (52.9%) | 48 (47.1%) | | 2.14 [1.28 ; 3.57] | | 2.41 [1.21 ; 4.81] |
| | Level 3 | 128 | 56 (43.8%) | 72 (56.3%) | | 3.10 [1.91 ; 5.02] | | 2.75 [1.38 ; 5.46] |
| | Level 4-5 | 40 | 15 (37.5%) | 25 (62.5%) | | 4.01 [1.95 ; 8.26] | | 2.89 [1.14 ; 7.34] |

* All variables with a univariate p-value < = 0.20 and with less than 20% of missing data were selected for the multivariate analysis

**"Stepwise" method with an entry threshold = 0.20% and an exit threshold = 0.05%

**Table 5. Characteristics of patients suffering from AR and Asthma.**

| | | Well controlled | Partly controlled | Poorly controlled | P-value [f] |
|---|---|---|---|---|---|
| | | N = 238 | N = 106 | N = 65 | |
| Time to AR diagnosis | (years) | 3.1 ± 5.3 | 4.4 ± 7.6 | 4.6 ± 6.9 | |
| Score NOSE (0-100) | | 49.9 ± 26.5 | 57.1 ± 24.4 | 58.2 ± 28.8 | 0.009 |
| n (%) | >50 | 110 (46.6%) | 63 (59.4%) | 38 (58.5%) | |
| | ≤50 | 126 (53.4%) | 43 (40.6%) | 27 (41.5%) | |
| RHINOQOL - Frequency (0-100) | | 64.4 ± 21.0 | 59.9 ± 18.4 | 62.5 ± 25.5 | 0.191 |
| RHINOQOL - Bothersomeness (0-100) | | 66.2 ± 22.9 | 62.9 ± 20.3 | 67.4 ± 22.4 | 0.284 |
| RHINOQOL - Impact (0-100) | | 23.1 ± 20.3 | 28.0 ± 20.2 | 29.8 ± 23.5 | 0.026 |
| DYNACHRON score (0-130) | | 44.8 ± 31.7 | 51.9 ± 29.5 | 55.9 ± 33.4 | |
| Score ACQ-6 | | 1.0 ± 0.9 | 1.6 ± 1.0 | 2.4 ± 1.2 | <0.001 |

[f]: Kruskal-Wallis test

suggests that AR flare-ups may also be a therapeutic target in AR, an indicator of AR control as a complement to ARIA's severity-based approach [12].

Our study has some limitations. We were expecting 1,768 to 2,079 patients in the total population. Potential biases were related to participating physician, patient selection and the patient's memory. Nguyen *et al* showed previously that most patients tended to grade symptoms as more severe before consultation showing that self-reported scores should be interpreted with caution, taking into account possible factors which may cause bias [30]. However, our results are consistent with other studies [28] as previously stated and provide original data to be confirmed by prospective cohort studies. Moreover, certain symptoms of AR perceived by patients may have been confused with the one from upper respiratory tract infection although apart from nasal obstruction and rhinorrhea they are different [31, 32].

However, our study was based on real-life conditions and on a large number of patients which permitted analysis of subgroups.

## Conclusion

This study showed that three quarters of patients had experienced at least one flare-up in the previous year. Polysensitization, severity of rhinitis and NOSE score >50 were strong indicators of uncontrolled AR. These results could be used to prioritize patients requiring increased treatment, notably earlier desensitization.

## Supporting information

**S1 File.**
(DOCX)

## Acknowledgments

**We would like to acknowledge all investigators involved in the study**:

A. Aferiat Derome (Marseille), M. Agell Perello (Hyères), S. Argoullon (Le Plessis Robinson), B. Atlani Boisseau (Cannes), P. Attal (Garges Les Gonesse), D. Aubry (Vannes), P. Azerad (Cannes), C. Barrière Tournier (Nimes), M. Beau Besnard (Le Bouscat), I. Begon Bagdassarian (Paris), A. Bentaleb (Amiens), A. Bernede Astruc (Lyon), C. Bertin (Limoges), V. Bertrac Valentin (Brest), V. Boisserie Lacroix (Lormont), JC. Bonneau (Angers), B. Bonnefoy Guionnet (St Lo), F. Bonte (Bourges), F. Bouvier (Le Havre), F. Bouvier Taulelle (Nimes), A. Broué Chabbert (Tournefeuille), PM. Broussier (Bois Guillaume), F. Brugère Largilliere (Gentilly), JP. Cabanettes (Rodez), I. Cabon Boudard (Marseille), B. Capy Noel (Reims), V. Castro (Limoges), JL. Cerutti (Annecy), N. Chapuis (Rochefort), Y. Cheraitia (Venissieux Cedex), A. Claudel (St Jean De La Ruelle), M. Claussner Paulignan (Forbach), A. Cleenewerck Picard (Bordeaux), HB. Co Minh Trinh (Olivet), M. Colas (Caluire Et Cuire), P. Collineau (St Maur Des Fosses), M. Comte Chemin (Tarnos), JM. Cordier (Cluses), J. Corriger Ippolito (Ludres), J. Cottet (Bonneval), L. Courtois Delair (Rueil Malmaison), C. Crepin Boetsch (Delle), M. Dan (Mulhouse), M. Danielou (Laval), O. Davigny Boddaert (St Quentin), R. De Lageneste (Colomiers), P. Debove (Cornebarrieu), Mc. Delsaux (Gournay Sur Marne), E. Des (Cugnaux), JM. Devoisins (Cholet), S. Dilem (Orsay Cedex), M. Dron Gonzalvez (Martigues), B. Dubegny (Mayenne), Y. Dubreil (Nantes), O. Dupin (Lattes), G. Dupont André (Levallois Perret), F. Dupuy (Riom), O. El Turk (St Dizier), A. Elali (Sanary Sur Mer), V. Etienne Tena (Bastia), JP. Fallot (Nimes), JL. Fontaine (Montauban), A. Gaillard Lopez (Ales), C. Gallen (Narbonne), F. Gaouar (Bretigny Sur Orge), T. Gentina (Lille), HP. Ghighi (Montauban), E. Girodet (Bron), JC. Gonzalvez (Port De Bouc), S. Graba (Paris), I. Grozelier (Cholet), R. Grüss (Amilly), A.

Guibert Longeon (Voiron), S. Guillo (Argeles Sur Mer), JL. Hallet (Luneville), I. Henry Daubas (Toulon), M. Honarmand (Grenoble), D. Horeau Paqueron (Laval), M. Ibrahimi (Dijon), N. Jaques Thauvin (Orléans), S. Jarlot Chevaux (Nancy), C. Jean (Sedan), A. Juchet (Toulouse), AS. Kerjan (Guingamp), MP. Kraus (St Dizier), R. Lachaussée (Cannes), C. Lagrange (Belleville), H. Laize (Sceaux), C. Langlet (Bayonne), M. Larchevesque Perimony (Rouen), M. Larrousse (Toulon), F. Le Pape Brouillet (Maisons Laffitte), F. Louis Donguy (Nice), MA. Luigi Postigo (Villefranche De Lauragais), V. Lustgarten Grillot (Nice), C. Mardini (Ste Foy Les Lyon), B. Martinez (Albi), C. Merault (Rouen), I. Mercier (Paris), V. Mercier (Plaisance Du Touch), A. Merzouk (Drancy), A. Millière (Dijon), E. Monereau (Antibes), P. Moudiki (Pontault Combault), DP. Mounier (St Etienne), R. Nguyen (Savigny Sur Orge), D. Noblet (Bourg La Reine), M. Nouvelle (St Amand Les Eaux), B. O Hana (Paris), C. Ohayon Elfasci (Paris), MJ. Pascalet Guidon (Marseille), C. Pasquet Noualhaguet (Bois D Arcy), S. Pestre (St Nicolas De Port), QB. Pham (Limeil Brevannes), I. Picard Bastide (Meze), I. Pierron Damalix (St Pierre D'Albigny), M. Pipart (Champagnole), B. Poitevin (Bormes Les Mimosas), B. Ponti (Toulouse), P. Poulain (Chaville), N. Prezelin (Reze), E. Puillandre (La Teste De Buch), A. Radig (Vieux Thann), G. Rapidie (Bordeaux), C. Rauld Deissard (Dijon), L. Refabert (Paris), F. Riotte Flandrois (Le Peage De Roussillon), MN. Robberecht Riquet (Mons En Baroeul), V. Saez Mora (Nimes), P. Salletaz (Toulon), R. Sanchez (St Priest), M. Sanmiguel (Vanves), D. Schwender (Dijon), A. Segyo (Nevers), A. Sergent (Rosny Sous Bois), A. Seriat Gautier (Rognac), E. Sève (Fontainebleau), M. Seynave (Lille), P. Shqeif (Poitiers), R. Sillam (Grenoble), G. Simon (Tours), D. Sror (Lingolsheim), L. Stoimenova (Angers), P. Tardieux (St Quentin), M. Thetis Soulie (Creteil), E. Thomas (Chaumont), A. Verbert (St Jean De Luz), A. Vial Dupuy (Paris), TM. Vuong (Noisiel), V. Zambelli (Grenoble), C. Zenoun (Toulouse), M. Zouari (Etampes).

## Author Contributions

**Conceptualization:** Ludovic de Gabory, Sabine Amet, Annelore Le Maux, Jean-Pierre Meunier, Antoine Chartier, Cécile Chenivesse.

**Data curation:** Ludovic de Gabory, Sabine Amet, Annelore Le Maux, Jean-Pierre Meunier, Antoine Chartier, Cécile Chenivesse.

**Formal analysis:** Ludovic de Gabory, Sabine Amet, Annelore Le Maux, Jean-Pierre Meunier, Antoine Chartier, Cécile Chenivesse.

**Funding acquisition:** Sabine Amet, Annelore Le Maux, Antoine Chartier.

**Methodology:** Ludovic de Gabory, Sabine Amet, Annelore Le Maux, Jean-Pierre Meunier, Antoine Chartier, Cécile Chenivesse.

**Project administration:** Sabine Amet, Annelore Le Maux, Jean-Pierre Meunier, Antoine Chartier.

**Resources:** Jean-Pierre Meunier.

**Software:** Annelore Le Maux.

**Supervision:** Sabine Amet, Annelore Le Maux, Antoine Chartier.

**Validation:** Ludovic de Gabory, Sabine Amet, Annelore Le Maux, Jean-Pierre Meunier, Antoine Chartier, Cécile Chenivesse.

**Visualization:** Ludovic de Gabory, Sabine Amet, Annelore Le Maux, Jean-Pierre Meunier, Antoine Chartier, Cécile Chenivesse.

**Writing – original draft:** Ludovic de Gabory, Sabine Amet, Annelore Le Maux, Jean-Pierre Meunier, Antoine Chartier, Cécile Chenivesse.

**Writing – review & editing:** Ludovic de Gabory, Sabine Amet, Annelore Le Maux, Jean-Pierre Meunier, Antoine Chartier, Cécile Chenivesse.

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
