## [Decision Letter · Decision Letter 0]

20 Sep 2022

PONE-D-22-18348Allergic rhinitis flare-ups and associated airways phenotype in house dust mite sensitization: a cross-sectional prospective studyPLOS ONE

Dear Dr. AMET,

Thank you for submitting your manuscript to PLOS ONE. After careful consideration, we feel that it has merit but does not fully meet PLOS ONE’s publication criteria as it currently stands. Therefore, we invite you to submit a revised version of the manuscript that addresses the points raised during the review process.

We look forward to receiving your revised manuscript.

Kind regards,

Sethu Thakachy Subha, M.S

Academic Editor

PLOS ONE

Journal Requirements:

2. In your statement, please include the full name of the IRB or ethics committee who approved or waived your study, as well as whether or not you obtained informed written or verbal consent. If consent was waived for your study, please include this information in your statement as well.

3. PLOS requires an ORCID iD for the corresponding author in Editorial Manager on papers submitted after December 6th, 2016. Please ensure that you have an ORCID iD and that it is validated in Editorial Manager. To do this, go to ‘Update my Information’ (in the upper left-hand corner of the main menu), and click on the Fetch/Validate link next to the ORCID field. This will take you to the ORCID site and allow you to create a new iD or authenticate a pre-existing iD in Editorial Manager. Please see the following video for instructions on linking an ORCID iD to your Editorial Manager account: https://www.youtube.com/watch?v=_xcclfuvtxQ.

6. We note that the grant information you provided in the ‘Funding Information’ and ‘Financial Disclosure’ sections do not match.

7. Thank you for stating the following financial disclosure:

“The study was funded by ALK SAS, France, who delegated operational management of the study to a contract research organization (Axonal-biostatem, Nanterre, France). ”

8. Please expand the acronym “ALK SAS” (as indicated in your financial disclosure) so that it states the name of your funders in full.

9. Thank you for stating the following in the Acknowledgments Section of your manuscript:

“The study was funded by ALK-Abelló S.A, France, who delegated operational management of the study to a contract research organization (Axonal-biostatem, Nanterre, France).

We would like to acknowledge Marjorie Lemonnier and Nicolas Lemaire (Axonal-Biostatem, Nanterre, France) for the management of the study and the statistical analyses; Françoise Bonnici (IciBio, Baden, France) for medical writing and editorial assistance and IPAC La Traduction de l'Industrie Pharmaceutique for English review.”

“The study was funded by ALK SAS, France, who delegated operational management of the study to a contract research organization (Axonal-biostatem, Nanterre, France).”

Reviewers' comments:

Reviewer's Responses to Questions

**Comments to the Author**

1. Is the manuscript technically sound, and do the data support the conclusions?

Reviewer #1: Partly

Reviewer #2: Yes

2. Has the statistical analysis been performed appropriately and rigorously? 

Reviewer #1: Yes

Reviewer #2: Yes

3. Have the authors made all data underlying the findings in their manuscript fully available?

Reviewer #1: Yes

Reviewer #2: Yes

4. Is the manuscript presented in an intelligible fashion and written in standard English?

Reviewer #1: Yes

Reviewer #2: Yes

5. Review Comments to the Author

Reviewer #1: In this study, the authors described the characteristics of flare-ups among 1701 HDM-AR patients. They found the occurrence of at least one AR flare-up in the year was 77.7% and the occurrence of equal or more than 2 AR flare-ups was 54.5%. They also described the variables related to frequently flare-up subgroups. They concluded that flare-ups were frequent and impaired QoL in HDM allergic patients. In general, this study is of some interests to readers as the flare-up is difficult to quantify in clinical practice. However, there are some questions to be addressed.

Major comments:

1. The definition of “flare-up” is not well clarified in this study. The authors stated “had any symptoms that were so severe that they required a change in their usual treatment and had an impact on their quality of life” in the methods. These criteria were obscure and might be difficult to duplicate in future studies.

2. The span of the study was one year, which was relatively long period. It would be difficult for the patients or their parents to recall the details of AR flare-ups, including symptoms, episodes of worsen, duration, trigger factors and other information. What actions did the authors taken to avoid or minimize the recall bias?

Minor comments:

1. How did the authors define the flare-up of AR in younger age patients?

2. Please check the numbers in Table 1 carefully, some data didn’t match well.

3. The definition of “AR exacerbation” should be noted in the text? Did it mean “flare-up”?

4. Why the symptom of itchy nose was not included in AR exacerbation in Table III

5. Could the author explain why some patients’ QoL were unchanged or even improved during AR flare-up? It was not in accordance with the definition of “flare-up” in this study.

Reviewer #2: The presented manuscript is of interest because of a very large group involved. The amount of data is impressive, but most of the conclusions are already known, for example, factors contributing to the allergic rhinitis flare-ups (whatever we call it).

Methodologically, thestudy is very well done - I have no comments.

The reader is interested in what happened next with the patients.....

Comments

1. Please write that the work was retrospective, possibly retrospective-prospective, in my opinion it did not meet the criteria of a prospective study.

2. SLIT-drops sounds confusing here because we do not know the specific preparation but we are guessing what it is. proposes to stay with AIT

3. please update the literature in the discussion

6. PLOS authors have the option to publish the peer review history of their article (what does this mean?). If published, this will include your full peer review and any attached files.

Reviewer #1: No

Reviewer #2: No

---

## [Author Response · Author response to Decision Letter 0]

7 Dec 2022

Allergic rhinitis flare-ups and associated airways phenotype in house dust mite sensitization: a cross-sectional prospective study

Dear editor, 

First, we would like to thank the reviewers and editor for their comments that have allowed us to improve the manuscript. Please find below our responses: 

Reviewer #1: In this study, the authors described the characteristics of flare-ups among 1701 HDM-AR patients. They found the occurrence of at least one AR flare-up in the year was 77.7% and the occurrence of equal or more than 2 AR flare-ups was 54.5%. They also described the variables related to frequently flare-up subgroups. They concluded that flare-ups were frequent and impaired QoL in HDM allergic patients. In general, this study is of some interests to readers as the flare-up is difficult to quantify in clinical practice. However, there are some questions to be addressed.

We thank the reviewer for this comment as well as the reviewing work that helped us improve the manuscript.

Major comments:

1. The definition of “flare-up” is not well clarified in this study. The authors stated “had any symptoms that were so severe that they required a change in their usual treatment and had an impact on their quality of life” in the methods. These criteria were obscure and might be difficult to duplicate in future studies.

We thank the reviewer for this very important point. AR flare-up is clinically defined as an acute and transient worsening of preexisting symptoms. It is difficult to give a precise definition because symptoms are not objectively measurable (Krzych-Fałta E, Samoliński BK. Objectification of the nasal patency assessment techniques used in nasal allergen provocation testing. Postepy Dermatol Alergol. 2020 Oct;37(5):635-640; Ta NH, Gao J, Philpott C. A systematic review to examine the relationship between objective and patient-reported outcome measures in sinonasal disorders: recommendations for use in research and clinical practice. Int Forum Allergy Rhinol. 2021 May;11(5):910-923), in particular nasal obstruction cannot be assessed unlike bronchial obstruction (spirometry). It should be noted, however, that patients know their symptoms, which are consistently similar during all flare-ups and that this symptomatology is different from that of an Upper Respiratory Tract infection (Elliott J. Whitaker M, Bodinier B, et al. Predictive symptoms for COVID-19 in the community: REACT-1 study of over 1 million people. PloS Med. 2021;18(9): e1003777; Czubak J, Stolarczyk K, Orzeł A, Frączek M, Zatoński T. Comparison of the clinical differences between COVID-19, SARS, influenza, and the common cold: A systematic literature review. Adv Clin Exp Med. 2021 Jan;30(1):109-114). Nowadays, there is no consensus on how to identify AR flare-ups in clinical practice. In the more general field of chronic rhinosinusitis, authors usually define flare-up based on use of systemic therapy, such as antibiotics or steroids, plans for surgical intervention, emergency department or urgent care visit, or hospitalization. However, by similarity to the field of asthma, these definitions refer to severe flare-ups, but let's not forget that severe allergic rhinitis does not lead to hospitalization and is not life-threatening like asthma. In our study, we aimed to capture all flare-ups whatever their severity and have therefore made the choice not to use such criteria. A recent systematic review published by Dawei Wu et al (Wu D, Bleier B, Wei Y. Definition and characteristics of acute exacerbation in adult patients with chronic rhinosinusitis: a systematic review. J Otolaryngol Head Neck Surg. 2020 Aug 18;49(1):62) found that the common point of all definitions used for AR flare-ups was the worsening of chronic rhinosinusitis symptoms with a return to baseline symptoms following treatment. In order to facilitate recall and overcome the daily variation of symptoms, we added the notion of impaired quality of life and the need to modify usual treatment. We agree that this definition has its own limits. However, our results are consistent with a recent publication by Jason H Kwah and al (Kwah JH, Somani SN, Stevens WW, Kern RC, Smith SS, Welch KC, Conley DB, Tan BK, Grammer LC, Yang A, Schleimer RP, Peters AT. Clinical factors associated with acute exacerbations of chronic rhinosinusitis. J Allergy Clin Immunol. 2020 Jun;145(6):1598-1605) that found 19.3% of patients with chronic rhinosinusitis experiencing more than 4 flare-ups requiring antibiotic per year based on a data collected from an electronical medical record (20.5% in our study). We developed this limit in the discussion of the revised manuscript.

2. The span of the study was one year, which was relatively long period. It would be difficult for the patients or their parents to recall the details of AR flare-ups, including symptoms, episodes of worsen, duration, trigger factors and other information. What actions did the authors taken to avoid or minimize the recall bias?

We agree with this remark. This is a common issue in airway disease studies that use the exacerbation rate as an endpoint. We could not find any data in allergic rhinitis but this strategy is usually applied in asthma and chronic obstructive pulmonary disease clinical trials. Moreover, we would like to point out that each patient has an AR flare-up phenotype which is always pretty much the same and they thus give a general description of their flare-up profile. The fact that our results are consistent with other studies conducted in the fairly closed field of chronic rhinosinusitis, at least in terms of frequency (Kwah JH, Somani SN, Stevens WW, Kern RC, Smith SS, Welch KC, Conley DB, Tan BK, Grammer LC, Yang A, Schleimer RP, Peters AT. Clinical factors associated with acute exacerbations of chronic rhinosinusitis. J Allergy Clin Immunol. 2020 Jun;145(6):1598-1605) makes us confident in our results. Although we agree that the description of AR flare-ups contains a certain degree of imprecision and must be confirmed by prospective studies. Our study provides original data which can constitute the basis to build prospective cohort studies. We have added these limits and perspectives in the revised manuscript.

Minor comments:

1. How did the authors define the flare-up of AR in younger age patients?

AR flare-up was defined in younger age patients in the same way as in adults but in this case, the flare-up count and description were reported by the parents. This has been clarified in the revised version of the manuscript.

2. Please check the numbers in Table 1 carefully, some data didn’t match well.

Thank you for your vigilance. We have checked all the numbers in Table 1 and adapted the manuscript accordingly.

3. The definition of “AR exacerbation” should be noted in the text? Did it mean “flare-up”?

Indeed, AR exacerbation and AR flare-up are synonyms. In order to homogenize the text, we have removed the term “AR exacerbation” and replaced it by “AR flare-up” throughout the manuscript.

4. Why the symptom of itchy nose was not included in AR exacerbation in Table III

We did not collect this data because itchy nose is inconstant and never isolated during flare-up of perennial allergic rhinitis, which is the case in house dust mite AR. Moreover, it’s not a major determinant of quality-of-life impairment in AR compared to nasal obstruction (cause of sleep disorders) or rhinorrhea and it’s not a major determinant of work productivity impairment. (Colás C, Galera H, Añibarro B, Soler R, Navarro A, Jáuregui I, Peláez A. Disease severity impairs sleep quality in allergic rhinitis (The SOMNIAAR study). Clin Exp Allergy. 2012 Jul;42(7):1080-7; Szeinbach SL, Seoane-Vazquez EC, Beyer A, Williams PB. The impact of allergic rhinitis on work productivity. Prim Care Respir J. 2007 Apr;16(2):98-105.) 

5. Could the author explain why some patients’ QoL were unchanged or even improved during AR flare-up? It was not in accordance with the definition of “flare-up” in this study.

Thank you for highlighting this point. Indeed, from a general point of view, allergic rhinitis is not the only determinant of the quality of life and the causal relationship is partly determined by the intensity of the disease. The number of flare-ups was collected using the question “In the past 12 months, have you had such severe symptoms ("exacerbation") that they required a change in your usual treatment and had an impact on your quality of life (daily activity, sleep, etc.)?” from the patient self-administered questionnaire. In order to assess the reliability of the answers, two additional questions were asked, taking up the determinants of the definition of exacerbation, one concerning the impact on quality of life and one concerning the use of an additional drug. According to our definition of flare-ups, we have added in the Table III the flare-ups requiring an additional treatment. This data was missing in the manuscript. 

However, the validated self-questionnaires of course all have their limits in terms of understanding the questions and the quality of the answers provided, explaining why there may be discrepancies. We must also not forget that allergic rhinitis is not the only determinant of the quality of life in a patient, which can be multifactorial: the family, professional context, physical, mental and psychological solidity and, conversely, the possibility of somatizing an ancillary problem towards the Otorhinolaryngological universe and to make a cause and effect relationship. Consequently, the discrepancy can go in both directions, but this study gives a trend in agreement with all the literature concerning the quality of life and the intensity of allergic rhinitis (Colás C, Galera H, Añibarro B, Soler R, Navarro A, Jáuregui I, Peláez A. Disease severity impairs sleep quality in allergic rhinitis (The SOMNIAAR study). Clin Exp Allergy. 2012 Jul;42(7):1080-7.). A prospective study could provide a more precise figure for exacerbation and a relationship with quality of life, without forgetting that the only measuring instrument currently available is self-administered questionnaires.

Reviewer #2: The presented manuscript is of interest because of a very large group involved. The amount of data is impressive, but most of the conclusions are already known, for example, factors contributing to the allergic rhinitis flare-ups (whatever we call it). Methodologically, the study is very well done - I have no comments. The reader is interested in what happened next with the patients.....

We thank the reviewer for this positive comment as well as the reviewing work that helped us improve the manuscript.

Comments

1. Please write that the work was retrospective, possibly retrospective-prospective, in my opinion it did not meet the criteria of a prospective study.

We agree that the wording is confusing. The best term qualifying our study design is probably “cross-sectional”. We firstly called it a prospective study as we collected data in real-time at the time of inclusion even if some of the data referred to events that occurred in the past. We don’t think that we can qualify this study as retrospective insofar as we have not collected existing data previously recorded for reasons other than the study. In order to avoid any confusion, we propose to remove the term “prospective” and to keep the term “cross-sectional”. This change has been made to the revised manuscript.

2. SLIT-drops sounds confusing here because we do not know the specific preparation but we are guessing what it is. proposes to stay with AIT

You are right, the term is confusing and we have homogenized the text with the term AIT.

3. please update the literature in the discussion

We have updated the literature. We have added 7 references between 2020 and 2021. 

On proofreading the manuscript, we identified some typos that we corrected. 

I hope the modification and answers will prove suitable and I remain at your disposal for any further information you may require. 

Sincerely yours, 

Pr Ludovic de Gabory

---

## [Decision Letter · Decision Letter 1]

6 Mar 2023

Cross-sectional study to describe allergic rhinitis flare-ups and associated airways phenotype in house dust mite sensitization.

PONE-D-22-18348R1

Dear Dr. AMET,

We’re pleased to inform you that your manuscript has been judged scientifically suitable for publication and will be formally accepted for publication once it meets all outstanding technical requirements.

Kind regards,

Sethu Thakachy Subha, M.S

Academic Editor

PLOS ONE

Additional Editor Comments (optional):

Reviewers' comments:

Reviewer's Responses to Questions

**Comments to the Author**

1. If the authors have adequately addressed your comments raised in a previous round of review and you feel that this manuscript is now acceptable for publication, you may indicate that here to bypass the “Comments to the Author” section, enter your conflict of interest statement in the “Confidential to Editor” section, and submit your "Accept" recommendation.

Reviewer #1: All comments have been addressed

Reviewer #3: All comments have been addressed

2. Is the manuscript technically sound, and do the data support the conclusions?

Reviewer #1: Yes

Reviewer #3: Yes

3. Has the statistical analysis been performed appropriately and rigorously? 

Reviewer #1: Yes

Reviewer #3: Yes

4. Have the authors made all data underlying the findings in their manuscript fully available?

Reviewer #1: Yes

Reviewer #3: Yes

5. Is the manuscript presented in an intelligible fashion and written in standard English?

Reviewer #1: Yes

Reviewer #3: Yes

6. Review Comments to the Author

Reviewer #1: The authors have addressed my queries adequately, I have no further questions. I think this paper is suitable for publication.

Reviewer #3: This observational cross-sectional study, utilizing accepted clinical tools for measuring patient outcomes, provides meaningful clinical information on patients including patient characteristics associated with flares, duration and severity of disease, and QOL. As noted by the reviewers, the data on severity and persistence is impacted by looser definitions; however, the authors' attempt to address these limitations are reasonable, and importantly the manuscript adds meaningfully to the literature. Furthermore, as cited in the discussion section, this paper should serve as a valuable resource for the design of future studies focused on patients with allergic rhinitis, allergic rhinosinusitis, and allergic rhinitis as a component of asthma (and the impact of these co-morbidities on asthma flares).

7. PLOS authors have the option to publish the peer review history of their article (what does this mean?). If published, this will include your full peer review and any attached files.

Reviewer #1: **Yes: **Rongfei Zhu

Reviewer #3: No

---

## [Editor Report · Acceptance letter]

14 Mar 2023

PONE-D-22-18348R1 

Cross-sectional study to describe allergic rhinitis flare-ups and associated airways phenotype in house dust mite sensitization. 

Dear Dr. Amet:

I'm pleased to inform you that your manuscript has been deemed suitable for publication in PLOS ONE. Congratulations! Your manuscript is now with our production department. 

Kind regards, 

on behalf of

Dr. Sethu Thakachy Subha 

Academic Editor

PLOS ONE